# *k*-Level Extended Sparse Array Design for Direction-of-Arrival Estimation

**Pinjiao Zhao [1,2,\*], Qisong Wu [2,\*], Na Wu [3], Guobing Hu [1] and Liwei Wang [4]**

[1] College of Electronic and Information Engineering, Jinling Institute of Technology, Nanjing 211169, China
[2] Key Laboratory of Underwater Acoustic Signal Processing of Ministry of Education, Southeast University, Nanjing 210096, China
[3] The School of Internet of Things, Nanjing University of Posts and Telecommunications, Nanjing 210003, China
[4] Nanjing Electronic Devices Institute, Nanjing 210007, China
\* Correspondence: zhaopinjiao@hrbeu.edu.cn (P.Z.); qisong.wu@seu.edu.cn (Q.W.);
   Tel.: +86-185-5183-6158 (Q.W.)

**Abstract:** Sparse arrays based on the concept of a sum-difference coarray (SDCA) have increased degrees of freedom and enlarged effective array aperture compared to those only considering a difference coarray. Nevertheless, there still exist a number of overlapping virtual sensors between the difference coarray and the sum coarray, yielding high coarray redundancy. In this paper, we propose a *k*-level extended sparse array configuration consisting of one sparse subarray with *k*-level expansion and one uniform linear subarray. By systematically analyzing the inherent structure of the *k*-level extended sparse array, the closed-form expressions for sensor locations, uniform DOF and coarray redundancy ratio (CARR) are derived. Moreover, with the utilization of a *k*-level extended strategy, the proposed array remains a hole-free property and achieves low coarray redundancy. According to the proposed sparse array, the spatial and temporal information of the incident sources are jointly exploited for underdetermined direction-of-arrival estimation. The theoretical propositions are proven and numerical simulations are performed to demonstrate the superior performance of the proposed array.

**Keywords:** sparse array; sum-difference coarray; degrees of freedom; coarray redundancy ratio; direction-of-arrival estimation

## 1. Introduction

Direction-of-arrival (DOA) estimation has been a core topic in various array signal processing applications, such as radar, sonar, wireless communications and electronic surveillance [1–6], where the sensor array plays a vital role in collecting the spatial sampling of impinging sources [7]. In contrast to the widely utilized uniform linear arrays (ULAs) [8,9], the emerging sparse arrays [10–12] have broken the limit of spatial sampling and exhibit remarkable properties in array flexibility, effective array aperture, and degrees of freedom (DOF) [13,14], and can thus handle the underdetermined scenarios wherein the number of incident sources exceeds that of physical sensors.

Benefiting from these advantages, a number of sparse array geometry designs have been investigated. The minimum redundancy array (MRA) [15] and the minimum hole array (MHA) [16] are the two typical ones, which, however, have no closed-form expressions for the array configurations. In contrast, the nested array (NA) [17] and coprime array (CPA) [18] can be systematically designed and the sensor locations can be expressed analytically. Typically, a two-level NA comprises two concatenated uniform linear subarrays with different spacings, which have a hole-free difference coarray (DCA) and can provide $\mathcal{O}(N^2)$ DOF with $\mathcal{O}(N)$ physical sensors. Based on the prototype of NA, several modified versions have been developed by rearranging the sensor locations, such as super NA [19], augmented NA [20], enhanced NA [21], and coprime NA [22], so as to further increase

the number of available DOF and alleviate the mutual coupling effects. Alternatively, a CPA is constructed with two coprime uniform linear subarrays and can provide $\mathcal{O}(MN)$ available DOF with $\mathcal{O}(M + N)$ physical sensors, whereas there exist holes in the DCA. To counter this, several variants including a coprime array with compressed inter-element spacing (CACIS) [23], coprime array with displaced subarrays (CADiS) [23] and thinned CPA [24] were developed. Nevertheless, all the aforementioned sparse array configurations accomplish DOA estimates using only the spatial information of impinging sources from the perspective of DCA equivalence, neglecting the corresponding temporal information.

Recently, several array designs based on sum-difference coarray (SDCA) have attracted considerable interest [25–28], wherein the spatial and temporal information of the received data can be fully utilized for DOA estimation. The SDCA is defined as the combination of sum co-array (SCA) and DCA, which has the potential to increase the number of DOF and resolve more sources. The prototype CPA [25] is improved based on SDCA, where more DOF than twice of the array aperture can be achieved. However, there exist several holes in the coarray and only the consecutive ones (also named as uniform DOF) are available for DOA estimation. In [26], a diff-sum coprime array with multiperiod subarrays (DsCAMpS) is designed to further enlarging the number of consecutive DOF. Similarly, two improved NAs [27] (i.e., INAwSDCA-I and INAwSDCA-II) have been presented by translating and flipping the subarrays of NA. However, for the aforementioned array configurations based on SDCA, lots of overlapping virtual sensors between their difference sets and sum sets lead to heavy coarray redundancy. Although some works, for example, the transformed nested arrays [28] (including TNA-I and TNA-II), were performed to reduce the number of redundant sensors, their coarrays still suffer from the existence of holes.

In this paper, a *k*-level extended sparse array configuration is designed with the union of one sparse subarray related to the extended coefficient *k* and one uniform linear subarray. It has simple closed-form expressions for array geometry and can achieve hole-free SDCA for any value of *k*. Moreover, the *k*-level extended strategy can further reduce the number of overlapping virtual sensors. Based on the proposed sparse array, the spatial and temporal information of incident sources are jointly exploited for DOA estimation. To be more specific, the main contributions of this work are summarized as follows:

(1)  We propose a novel sparse array configuration based on the concept of SDCA, termed as a *k*-level extended sparse array, which has an increased uniform DOF and reduced coarray redundancy simultaneously.

(2)  The closed-form expressions for the array configuration and uniform DOF are derived, and the hole-free property of the SDCA is analyzed and proven.

(3)  The coarray redundancy ratio of the *k*-level extended sparse array is derived for evaluating the coarray redundancy between DCA and SCA quantitatively.

Notations: we use lower-case letters, bold letters and upper-case bold letters to characterize scalars, vectors and matrices, respectively. $(\cdot)^{\mathrm{T}}$, $(\cdot)^{\mathrm{H}}$, and $(\cdot)^{*}$ and $\mathrm{E}[\cdot]$ represent the transpose, conjugate transpose, conjugate, and the statistical expectations, respectively. $\mathbf{I}_N$ is an $N \times N$ identity matrix and $\mathbb{Z}^{+}$ is the positive integer set. Additionally, $\mathrm{vec}(\cdot)$, $\mathrm{diag}(\cdot)$ and $\mathrm{card}(\cdot)$ are, respectively, the vectorization operator, diagonalization operator and cardinality operator. The symbol $\otimes$ denotes the Kronecker product, and $\odot$ denotes the Khatri–Rao product. $\partial f(n)/\partial n$ denotes the partial derivative of function $f(n)$ to $n$.

## 2. Preliminaries

### 2.1. Signal Model

Consider a sparse array with $N$ sensors located at the positions $\overline{\mathbb{P}} = \mathbb{P} \times d = \{p_i | p_i \in \mathbb{Z}, i = 1, 2, \cdots, N\} \times d$, where $d = \lambda/2$ is the fundamental spacing with $\lambda$ being the signal wavelength that can be normalized hereinafter. Without loss of generality, the sensor locations in $\overline{\mathbb{P}}$ are arranged in ascending order, such that $p_i < p_{i+1}$ holds for $i = 1, 2, \cdots, N - 1$. Assume that $K$ far-field narrowband sources from directions

$\{\theta_1, \theta_2, \cdots, \theta_K | \theta_k \in [-\pi/2, \pi/2], k = 1, 2, \cdots, K\}$ with respective powers $\{\sigma_1^2, \sigma_2^2, \cdots, \sigma_K^2\}$ impinge on the sparse array. Then, the received data at time $t$ can be modeled as

$$\mathbf{x}(t) = \mathbf{A}\mathbf{s}(t) + \mathbf{n}(t) = \sum_{k=1}^{K} \mathbf{a}(\theta_k)s_k(t) + \mathbf{n}(t), \tag{1}$$

where $\mathbf{A} = [\mathbf{a}(\theta_1), \mathbf{a}(\theta_2), \cdots, \mathbf{a}(\theta_K)]$ is the array manifold matrix with $\mathbf{a}(\theta_k) = \left[e^{j\pi p_1 \sin\theta_k}, e^{j\pi p_2 \sin\theta_k}, \cdots, e^{j\pi p_N \sin\theta_k}\right]^{\mathrm{T}}$ being the steering vector corresponding to angle $\theta_k$. $\mathbf{s}(t) = [s_1(t), s_2(t), \cdots, s_K(t)]^{\mathrm{T}}$ denotes the source vector whose $k$th $(k = 1, 2, \cdots, K)$ column can be expressed as $s_k(t) = G_k e^{jw_k t}$ with $G_k$ and $w_k$ being the deterministic complex amplitude and frequency offset, respectively. $\mathbf{n}(t) = [n_1(t), n_2(t), \cdots, n_N(t)]$ is assumed to be the temporally and spatially Gaussian white noise vector that follows $N$-dimensional complex Gaussian distribution $\mathcal{CN}\left(0, \sigma_n^2 \mathbf{I}_N\right)$ and is independent from the source vector.

For convenience, several operations and terminologies used throughout this paper are defined as follows.

**Definition 1:** *For two given integer sets $\mathbb{S}_m$ and $\mathbb{S}_n$, four basic operations can be defined as:*
    *Self-difference operation: $\mathbb{D}_{sd} = \mathbb{S}_m - \mathbb{S}_m$, $m = 1, 2$.*
    *Cross-difference operation: $\mathbb{D}_{cd} = \mathbb{S}_m - \mathbb{S}_n$, $m, n = 1, 2$ and $m \neq n$.*
    *Self-sum operation: $\mathbb{D}_{ss} = \mathbb{S}_m + \mathbb{S}_m$, $m = 1, 2$.*
    *Cross-sum operation: $\mathbb{D}_{cs} = \mathbb{S}_m + \mathbb{S}_n$, $m, n = 1, 2$ and $m \neq n$.*

**Definition 2 (*Sum-Difference Coarray*):** *According to Definition 1, denote $\mathbb{P} = \{p_1, p_2, \cdots, p_N\}$ as the set of sensor indexes, the corresponding sum-difference coarray is defined as $\mathbb{D}_{sdca} = \{p_m - p_n\} \cup \{p_m + p_n\} \cup \{-p_m - p_n\} \, \forall p_m, p_n \in \mathbb{P}$.*

**Definition 3 (*Degrees of Freedom, DOF*):** *The number of degrees of freedom for a given sparse array configuration $\mathbb{P}$ is defined as the cardinality of its sum-difference coarray $\mathbb{D}_{sdca}$, i.e., $\mathrm{DOF}(\mathbb{P}) = \mathrm{card}(\mathbb{D}_{sdca})$.*

**Definition 4 (*Uniform DOF*):** *Denote $\mathbb{U}_{sdca}$ as the maximum consecutive segment of sum-difference coarray $\mathbb{D}_{sdca}$, then the uniform DOF of a sparse array configuration $\mathbb{P}$ is defined as the cardinality of $\mathbb{U}_{sdca}$, i.e., $uDOF(\mathbb{P}) = card(\mathbb{U}_{sdca})$, which is also known as the number of consecutive and unique lags in the sum-difference coarray.*

**Definition 5 (*Hole-free property/Restricted array*):** *If the DOF of a given sparse array $\mathbb{P}$ is equivalent to its uniform DOF, i.e., $uDOF(\mathbb{P}) = DOF(\mathbb{P})$, then the corresponding sum-difference coarray is said to be hole-free. As such, the sparse array $\mathbb{P}$ is called a restricted array.*

In view of the existing sparse array configurations, the self-difference operation and cross-difference operation are widely utilized in constructing the vectorized covariance matrix of received data, yielding a number of virtual sensors in the DCA domain. Based on this, the introduced sum coarray in the sum-difference coarray can further enlarge the number of virtual sensors from the perspective of coarray domain extension instead of array configuration design, which provide a new perspective for improving array performance and angle accuracy.

*2.2. DOA Estimation*

By collecting $T_s$ samples from the outputs of the $m$th sensor and $n$th sensor $\forall m, n \in [1, N], m \neq n$, denoted as $x_m(t)$ and $x_n(t)$, the time average function can be defined as

$$R_{x_m^* x_n}(\tau) = \frac{1}{T_s} \sum_{t=1}^{T_s} x_m^*(t) x_n(t + \tau) \approx \sum_{k=1}^{K} e^{j\pi(p_n - p_m)\sin\theta_k} R_{s_k^* s_k}(\tau) + R_{n_m^* n_n}(\tau), \tag{2}$$

where $\tau \neq 0$ denotes the time lag. Notice that $R_{s_k^* s_k}(\tau) = |G_k|^2 e^{j w_k t}$ has the same form as $s_k(t) = G_k e^{j w_k t}$, which can thus be treated as an equivalent signal with enlarged amplitude $|G_k|^2$ and invariant frequency offset $w_k$. Accordingly, $R_{x_m^* x_n}(\tau)$ can be seen as the equivalent received data of a generated virtual sensor whose location is $p_n - p_m$. In addition, the noise term of (2) can be removed according to $R_{n_m^* n_n}(\tau) = \sigma_n^2 \delta(n-m)\delta(\tau) = 0$, which implies that the noise component in this model can be suppressed. Without loss of generality, we chose the $m$th sensor for the reference, i.e., $m = 1$, then one can construct the time average vectors with lag $\tau$ and its mirrored version $-\tau$, as

$$\boldsymbol{\gamma}_x(\tau) = \mathbf{A}\boldsymbol{\gamma}_s(\tau), \boldsymbol{\gamma}_x(-\tau) = \mathbf{A}\boldsymbol{\gamma}_s(-\tau), \tag{3}$$

where $\boldsymbol{\gamma}_x(\tau) = \left[ R_{x_1^* x_1}(\tau), R_{x_1^* x_2}(\tau), \cdots, R_{x_1^* x_N}(\tau) \right]^{\mathrm{T}}$, $\boldsymbol{\gamma}_x(-\tau) = \left[ R_{x_1^* x_1}(-\tau), R_{x_1^* x_2}(-\tau), \cdots, \right.$ $\left. R_{x_1^* x_N}(-\tau) \right]^{\mathrm{T}}$, $\boldsymbol{\gamma}_s(\tau) = \left[ R_{s_1^* s_1}(\tau), R_{s_2^* s_2}(\tau), \cdots, R_{s_K^* s_K}(\tau) \right]^{\mathrm{T}}$, and $\boldsymbol{\gamma}_s(-\tau) = \left[ R_{s_1^* s_1}(-\tau), R_{s_2^* s_2}(-\tau), \right.$ $\left. \cdots, R_{s_K^* s_K}(-\tau) \right]^{\mathrm{T}}$. Combining $\boldsymbol{\gamma}_x(\tau)$ and $\boldsymbol{\gamma}_x(-\tau)$ yields a conjugate augmented vector

$$\boldsymbol{\gamma}(\tau) = \left[ \boldsymbol{\gamma}_x^{\mathrm{H}}(-\tau), \boldsymbol{\gamma}_x^{\mathrm{T}}(\tau) \right]^{\mathrm{T}} = \left[ \mathbf{A}^{\mathrm{H}}, \mathbf{A}^{\mathrm{T}} \right]^{\mathrm{T}} \boldsymbol{\gamma}_s(\tau), \tag{4}$$

By collecting $T_p$ pseudo snapshots, the pseudo-data matrix can be calculated by

$$\tilde{\boldsymbol{\gamma}} = \left[ \boldsymbol{\gamma}(P_s), \boldsymbol{\gamma}(2P_s), \cdots, \boldsymbol{\gamma}(T_p P_s) \right] = \left[ \mathbf{A}^{\mathrm{H}}, \mathbf{A}^{\mathrm{T}} \right]^{\mathrm{T}} \mathbf{G}\Theta, \tag{5}$$

where $T_p$ is the number of pseudo snapshots and $P_s$ is the pseudo sampling period satisfying the Nyguist sampling theorem. $\mathbf{G} = \mathrm{diag}\left( |G_1|^2, |G_2|^2, \cdots, |G_K|^2 \right)$, and $\Theta = \left[ \boldsymbol{\varphi}_1, \boldsymbol{\varphi}_2, \cdots, \boldsymbol{\varphi}_{T_p} \right]$ with the $n_p$ th column being $\boldsymbol{\varphi}_{n_p} = \left[ e^{j w_1 n_p P_s}, e^{j w_2 n_p P_s}, \cdots, e^{j w_K n_p P_s} \right]^{\mathrm{T}}$. Then, the covariance matrix of $\tilde{\boldsymbol{\gamma}}$ can be calculated as

$$\mathbf{Z}_{\gamma\gamma} = \mathrm{E}\left[ \tilde{\boldsymbol{\gamma}} \tilde{\boldsymbol{\gamma}}^{\mathrm{H}} \right] = \left[ \mathbf{A}^{\mathrm{H}}, \mathbf{A}^{\mathrm{T}} \right]^{\mathrm{T}} \mathbf{Z}_{ss} \left[ \mathbf{A}^{\mathrm{T}}, \mathbf{A}^{\mathrm{H}} \right] \approx \frac{1}{T_p} \sum_{n_p=1}^{T_p} \boldsymbol{\gamma}(n_p P_s) \boldsymbol{\gamma}^{\mathrm{H}}(n_p P_s), \tag{6}$$

where $\mathbf{Z}_{ss} = \mathbf{G}^2 = \mathrm{diag}\left( |G_1|^4, |G_2|^4, \cdots, |G_K|^4 \right)$. By vectorizing $\mathbf{Z}_{\gamma\gamma}$, we have

$$\mathbf{z}_{\gamma\gamma} = \mathrm{vec}(\mathbf{Z}_{\gamma\gamma}) = \left( \left[ \mathbf{A}^{\mathrm{H}}, \mathbf{A}^{\mathrm{T}} \right]^{\mathrm{H}} \odot \left[ \mathbf{A}^{\mathrm{H}}, \mathbf{A}^{\mathrm{T}} \right]^{\mathrm{T}} \right) \mathbf{z}_{ss}, \tag{7}$$

where $\mathbf{z}_{ss} = \mathrm{diag}(\mathbf{Z}_{ss}) = \left[ |G_1|^4, |G_2|^4, \cdots, |G_K|^4 \right]$ and the term $\left( \left[ \mathbf{A}^{\mathrm{H}}, \mathbf{A}^{\mathrm{T}} \right]^{\mathrm{H}} \odot \left[ \mathbf{A}^{\mathrm{H}}, \mathbf{A}^{\mathrm{T}} \right]^{\mathrm{T}} \right)$ behaves like a virtual array manifold matrix whose $k$th column can be expressed as

$$\tilde{\mathbf{a}}(\theta_k) = \begin{bmatrix} \mathbf{a}(\theta_k) \otimes \mathbf{a}^*(\theta_k) \\ \mathbf{a}(\theta_k) \otimes \mathbf{a}(\theta_k) \\ \mathbf{a}^*(\theta_k) \otimes \mathbf{a}^*(\theta_k) \\ \mathbf{a}^*(\theta_k) \otimes \mathbf{a}(\theta_k) \end{bmatrix}, \tag{8}$$

where the union of the virtual subarrays corresponding to $\mathbf{a}(\theta_k) \otimes \mathbf{a}^*(\theta_k) = \bigcup_{\forall p_{v1}, p_{v1} \in \mathbb{P}} \left\{ e^{j\pi(p_{v1} - p_{v2})\sin\theta_k} \right\}$ and $\mathbf{a}^*(\theta_k) \otimes \mathbf{a}(\theta_k) = \bigcup_{\forall p_{v1}, p_{v1} \in \mathbb{P}} \left\{ e^{j\pi(-p_{v1} + p_{v2})\sin\theta_k} \right\}$ is referred to as DCA, while the union of virtual subarrays corresponding to $\mathbf{a}(\theta_k) \otimes \mathbf{a}(\theta_k) = \bigcup_{\forall p_{v1}, p_{v1} \in \mathbb{P}} \left\{ e^{j\pi(p_{v1} + p_{v2})\sin\theta_k} \right\}$ and $\mathbf{a}^*(\theta_k) \otimes \mathbf{a}^*(\theta_k) = \bigcup_{\forall p_{v1}, p_{v1} \in \mathbb{P}} \left\{ e^{-j\pi(p_{v1} + p_{v2})\sin\theta_k} \right\}$ is named as SCA. Following that, the whole virtual array corresponding to $\tilde{\mathbf{a}}(\theta_k)$ is named as SDCA, which has been defined in Definition 2. As compared to the concept of DCA, SDCA is

defined as the combination of DCA and SCA, which has the potential to provide more DOF and larger array aperture, and more sources can be resolved accordingly. By deleting the repeated data and extracting the maximum continuous segment of the SDCA, spatial smoothing methods [29,30] or CS approaches [31,32] are employed for DOA estimation.

## 3. *k*-Level Extended Array Design

### 3.1. Motivations

Recently, the emerging sparse array configuration designs based on the concept of SDCA have attracted great attention [23–26], benefiting from the increasing DOF and the enlarged array aperture. Nevertheless, for most existing sparse arrays, such as NA (and its modified versions) [25,26] and CPA (and its modified versions) [23,24], lots of existing overlapping virtual sensors between their DCAs and SCAs lead to high coarray redundancy. Despite several works, such as TNA-I and TNA-II, have proposed to reduce the coarray redundancy, they have holes in their SDCAs. As an example, the DCAs, SCAs, and SDCAs of NA, CPA, DsCAMps, TNA-I, and TNA-II with six sensors are compared in Figure 1 with the sensor locations being $\mathbb{P}_{NA} = \{1, 2, 3, 4, 8, 12\}$, $\mathbb{P}_{CPA} = \{0, 3, 4, 6, 8, 9\}$, $\mathbb{P}_{DsCAMps} = \{0, 2, 3, 4, 6, 9\}$, $\mathbb{P}_{TNA-I} = \{0, 4, 8, 9, 10, 11\}$, and $\mathbb{P}_{TNA-II} = \{0, 4, 8, 10, 11, 13\}$, where black circles, white circles, and grey circles, respectively, denote DCA, consecutive parts of SCA, and the overlapping sensors. For illustrating clearly, the overlapping virtual sensors are marked by a red dashed circle box. The results from Figure 1 show that NA has 20 overlapping virtual sensors, CPA has 6 overlapping virtual sensors, DsCAMps has 14 overlapping virtual sensors, TNA-I has 8 overlapping virtual sensors, and TNA-II has 6 overlapping virtual sensors.

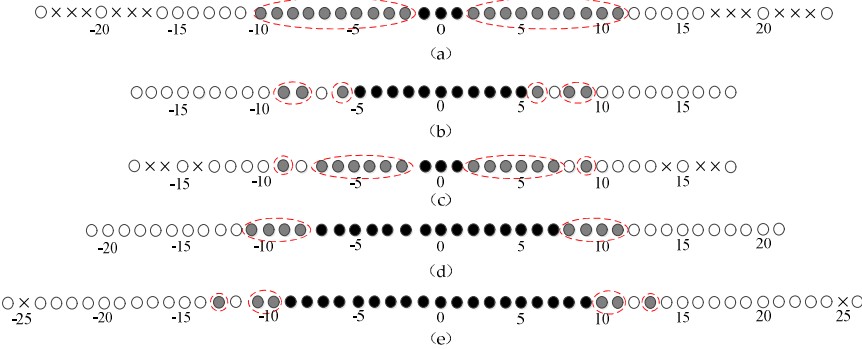

**Figure 1.** Comparisons of coarray distribution with $N$ = 6 sensors (**a**) NA; (**b**) CPA; (**c**) DsCAMps; (**d**) TNA-I; (**e**) TNA-II, where black circles, white circles, and grey circles are respectively denote DCA, consecutive parts of SCA, and the overlapping sensors, the overlapping virtual sensors are also marked by red dashed circle boxes.

Motivated by this, a novel array configuration is designed to further reduce the number of overlapping virtual sensors between their DCAs and SCAs, and the corresponding SDCAs maintain the hole-free property. As such, more uniform DOF can be utilized to improve DOA estimation accuracy and angular resolution. We name this sparse array configuration as the k-level extended sparse array and formally study it as follows.

### 3.2. Array Design Rules and Properties

**Definition 6:** *Assume that $2 < N_1 \leq N_2$ with $N_1 \in \mathbb{Z}^+$ and $N_2 \in \mathbb{Z}^+$, the k-level extended sparse array is defined as a union of one generated sparse subarray $\mathbb{P}_o^K$ by k-level expansion and one uniform linear subarray $\mathbb{P}_s$ with $N = kN_1 + N_2$ sensors, whose sensor locations are denoted as $\mathbb{P} = \mathbb{P}_o^K \cup \mathbb{P}_s$, where*

$$\begin{cases} \mathbb{P}_o^k = K_{\text{level}}(\mathbb{P}_o) \\ \mathbb{P}_o = \{1 + (p-1)N_1 | p \in [1, N_1]\}, \\ \mathbb{P}_s = \{k, N_1^2, +, p, | p \in [1, N_1]\} \end{cases} \qquad (9)$$

where $\mathbb{P}_o$ *is called a sparse base-subarray, $K_{level}(\cdot)$ is the k-level extended operator and k is the extended coefficient that can be any positive integer regardless of the values of $N_1$ and $N_2$. It is noted that the proposed k-level extended array configuration generally contains two subarrays: one subarray with $kN_1$ sensors generating from the k-level expansion of the sparse base-subarray and a uniform linear subarray with $N_2$ sensors. Accordingly, the array aperture of the proposed k-level extended sparse array is $kN_1^2 + N_2$. Based on the above array design rules, the proportion of larger interelement spacings would increase with the increase in k, such that the mutual effects among sensors can be alleviated.*

Then, we illustrate the structure of the proposed *k*-level extended sparse array by a specific example. Figure 2 shows the structures of the array configurations with respect to the values of *k* under the conditions of $N_1 = N_2 = 3$. Figure 2a plots the 1-level extended sparse array with six sensors whose locations are set to be $\{1, 4, 7, 10, 11, 12\}$. By comparison, two-level extended sparse array and three-level extended sparse array are, respectively, plotted in Figure 2b,c, where the former consists of a six-sensor sparse subarray and a three-sensor uniform linear subarray with sensor locations of $\{1, 4, 7, 10, 13, 16, 19, 20, 21\}$ and the latter contains a nine-sensor sparse subarray and three-sensor uniform linear subarray with sensor locations being $\{1, 4, 7, 10, 13, 16, 19, 22, 25, 28, 29, 30\}$. In addition, the proportion of sparse sensor number varies from 0.5 to 0.75 when the values of *k* ranging from 1 to 3. This indicates that the mutual effects can be alleviated with the increase in *k*.

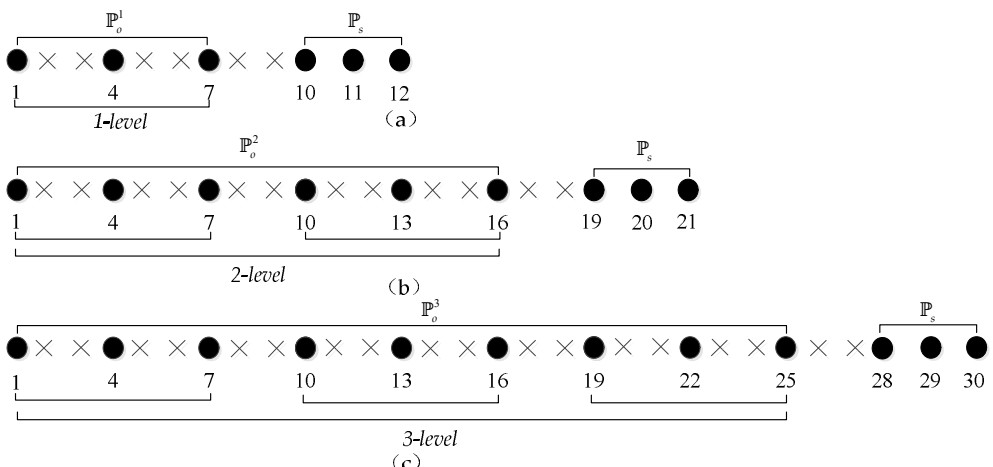

**Figure 2.** An example of *k*-level extended array with $N_1 = N_2 = 3$: (**a**) k = 1; (**b**) k = 2; (**c**) k = 3.

**Definition 7 (*Spatial Efficiency*):** *By extracting the positive parts of uniform DOF and the whole DOF in the sum-difference coarray, referred to as $uDOF^+$ and $DOF^+$, respectively, the spatial efficiency of a sparse array is defined as the ratio of $uDOF^+$ to $DOF^+$, i.e., $\eta = uDOF^+/DOF^+$.*

**Proposition 1.** *The spatial efficiency of the k-level extended sparse array is 1, which implies that the sum-difference coarray of the proposed array configuration has hole-free virtual ULAs.*

**Proof of Proposition 1.** For a given *k*-level extended array with $N = kN_1 + N_2$ sensors, since the SDCA is symmetric 0, the positive part of $\mathbb{D}_{sdca}^+$ can be constructed as

$$\mathbb{D}_{sdca}^+ = \underbrace{\mathbb{D}_{\mathbb{P}_o^k - \mathbb{P}_o^k} \cup \mathbb{D}_{\mathbb{P}_s - \mathbb{P}_s} \cup \mathbb{D}_{\mathbb{P}_s - \mathbb{P}_o^k}}_{\mathbb{D}_{dca}^+} \cup \underbrace{\mathbb{D}_{\mathbb{P}_o^k + \mathbb{P}_o^k} \cup \mathbb{D}_{\mathbb{P}_s + \mathbb{P}_s} \cup \mathbb{D}_{\mathbb{P}_o^k + \mathbb{P}_s}}_{\mathbb{D}_{sca}^+}, \tag{10}$$

where $\mathbb{D}_{\mathbb{P}_o^k - \mathbb{P}_o^k}, \mathbb{D}_{\mathbb{P}_s - \mathbb{P}_s}$ and $\mathbb{D}_{\mathbb{P}_s - \mathbb{P}_o^k}$ are the positive difference sets, $\mathbb{D}_{\mathbb{P}_o^k + \mathbb{P}_o^k}, \mathbb{D}_{\mathbb{P}_s + \mathbb{P}_s}$ and $\mathbb{D}_{\mathbb{P}_o^k + \mathbb{P}_s} \mathbb{D}_{\mathbb{P}_s - \mathbb{P}_o^k}$ are the positive sum sets, as

$$
\begin{aligned}
\mathbb{D}_{\mathbb{P}_o^k - \mathbb{P}_o^k} &= \left\{ p_o^k N_1 \middle| p_o^k \in \mathbb{Z} \text{ and } p_o^k \in \left[ 0, kN_1^2 - kN_1 \right] \right\} \\
\mathbb{D}_{\mathbb{P}_s - \mathbb{P}_s} &= \left\{ p_s | p_s \in \mathbb{Z} \text{ and } p_s \in [0, N_2 - 1] \right\} \\
\mathbb{D}_{\mathbb{P}_s - \mathbb{P}_o^k} &= \left\{ N_1, N_1 + 1, \cdots, kN_1^2 + N_2 - 2, kN_1^2 + N_2 - 1 \right\} \\
\mathbb{D}_{\mathbb{P}_o^k + \mathbb{P}_o^k} &= \left\{ 2, N_1 + 2, 2N_1 + 2, \cdots, 2kN_1^2 - 2N_1 + 2 \right\} \\
\mathbb{D}_{\mathbb{P}_s + \mathbb{P}_s} &= \left\{ 2kN_1^2 + p_s | p_s \in \mathbb{Z} \text{ and } p_s \in [2, 2N_2] \right\} \\
\mathbb{D}_{\mathbb{P}_o^k + \mathbb{P}_s} &= \left\{ kN_1^2 + 2, kN_1^2 + 3, \cdots, kN_1^2 + N_2 + 1, \cdots, 2kN_1^2 + N_2 - N_1 + 1 \right\}
\end{aligned}
\tag{11}
$$

For the positive difference sets, $\mathbb{D}_{\mathbb{P}_s - \mathbb{P}_o^k}$ is continuous in the range of $\left[ N_1, kN_1^2 + N_2 - 1 \right]$ and $\mathbb{D}_{\mathbb{P}_s - \mathbb{P}_s}$ is continuous in the range of $[0, N_2 - 1]$. Since $N_2 \geq N_1$ with $N_1 \in \mathbb{Z}^+$ and $N_2 \in \mathbb{Z}^+$, $\mathbb{D}_{\mathbb{P}_s - \mathbb{P}_s} \cup \mathbb{D}_{\mathbb{P}_s - \mathbb{P}_o^k} \subseteq \mathbb{D}_{dca}^+$ contains continuous lags in the range of $\left[ 0, kN_1^2 + N_2 - 1 \right]$. It is obvious that $\mathbb{D}_{\mathbb{P}_o^k - \mathbb{P}_o^k} \subseteq \mathbb{D}_{\mathbb{P}_s - \mathbb{P}_s} \cup \mathbb{D}_{\mathbb{P}_s - \mathbb{P}_o^k}$, thus the positive difference set $\mathbb{D}_{dca}^+$ is continuous in the range of $\left[ 0, kN_1^2 + N_2 - 1 \right]$. For the positive sum sets, $\mathbb{D}_{\mathbb{P}_o^k + \mathbb{P}_s}$ is continuous in the range of $\left[ kN_1^2 + 2, 2kN_1^2 + N_2 - N_1 + 1 \right]$ and $\mathbb{D}_{\mathbb{P}_s + \mathbb{P}_s}$ is continuous in the range of $\left[ 2kN_1^2 + 2, 2kN_1^2 + 2N_2 \right]$. Due to $N_2 \geq N_1$, $\mathbb{D}_{\mathbb{P}_o^k + \mathbb{P}_s}$ and $\mathbb{D}_{\mathbb{P}_s + \mathbb{P}_s}$ can be connected with no holes, ranging from $kN_1^2 + 2$ to $2kN_1^2 + 2N_2$. Note that $\mathbb{D}_{\mathbb{P}_o^k + \mathbb{P}_o^k}$ is composed of a series of scatters, satisfying $\mathbb{D}_{\mathbb{P}_o^k + \mathbb{P}_o^k} \subseteq \mathbb{D}_{\mathbb{P}_o^k + \mathbb{P}_s} \cup \mathbb{D}_{\mathbb{P}_s + \mathbb{P}_s}$, and thus the positive sum set $\mathbb{D}_{dca}^+$ is continuous in the range of $\left[ kN_1^2 + 2, 2kN_1^2 + 2N_2 \right]$. In terms of $2 < N_1 \leq N_2$ with $N_1 \in \mathbb{Z}^+$ and $N_2 \in \mathbb{Z}^+$, we have $kN_1^2 + N_2 - 1 - \left( kN_1^2 + 2 \right) \geq 0$, which implies that the positive DCA can fill all the holes in positive SCA and generate a hole-free positive SDCA in the range of $\left[ 0, 2kN_1^2 + 2N_2 \right]$. Therefore, the positive part of the uniform DOF is the same as that of the whole DOF, such that the spatial efficiency of the $k$-level extended sparse array is 1. According to symmetry, the whole SDCA of the proposed array configuration has hole-free virtual ULAs. $\square$

**Corollary 1.** *From the perspective of SDCA, the uniform DOF of the k-level extended sparse array is $4kN_1^2 + 4N_2 + 1$.*

**Proof of Corollary 1.** Based on Proposition 1, the $k$-level extended sparse array has hole-free DCA in the range of $\left[ -kN_1^2 - N_2 + 1, kN_1^2 + N_2 - 1 \right]$ and the consecutive range of the corresponding SCA is $\left[ -2kN_1^2 - 2N_2, -kN_1^2 - 2 \right] \cup \left[ kN_1^2 + 2, 2kN_1^2 + 2N_2 \right]$. Consequently, combining with DCA and the consecutive part of SCA, the generative SDCA has virtual ULAs in the range of $\left[ -2kN_1^2 - 2N_2, 2kN_1^2 + 2N_2 \right]$, which implies that the uniform DOF of the $k$-level extended sparse array is $4kN_1^2 + 4N_2 + 1$. $\square$

**Corollary 2.** *The optimal uniform DOF of the k-level extended sparse array is $4kN^2/(k+1)^2 + 4N/(k+1) + 1$ under the condition of $N_1 = N_2 = N/(k+1)$.*

**Proof of Corollary 2.** The above optimization problem can be constructed as

$$
\begin{aligned}
\max\ & uDOF = 4kN_1^2 + 4N_2 + 1 \\
& \text{s.t.} N = kN_1 + N_2
\end{aligned}
\tag{12}
$$

By calculating the partial derivative of $uDOF$ with respect to $N_1$, we have $\partial uDOF/\partial N_1 = 4k(2N_1 - 1) > 0$, which implies that the uniform DOF of the proposed array configuration would increase monotonically with the increase in $N_1$. Since $N_1 \leq N_2$, the maximum $uDOF$ can be obtained from $N_1 = N_2 = N/(k+1)$, and the corresponding optimal uniform DOF is calculated as $4kN^2/(k+1)^2 + 4N/(k+1) + 1$.

To further illustrate the coarray distribution of the $k$-level extended sparse array, the DCAs, consecutive parts of SCAs and generative SDCAs corresponding to the example

given in Figure 2 are plotted in Figure 3, where black circles, white circles, and grey circles, respectively, denote DCA, consecutive parts of SCA and the overlapping sensors. Since these coarrays are symmetric about 0, here the DCAs, consecutive SCAs, and SDCAs in Figure 3 refer to their non-negative parts. The results from Figure 3 show that there exists only one overlapping virtual sensor between DCA and SCA, which means almost all the unique lags contribute to increasing the uniform DOF. □

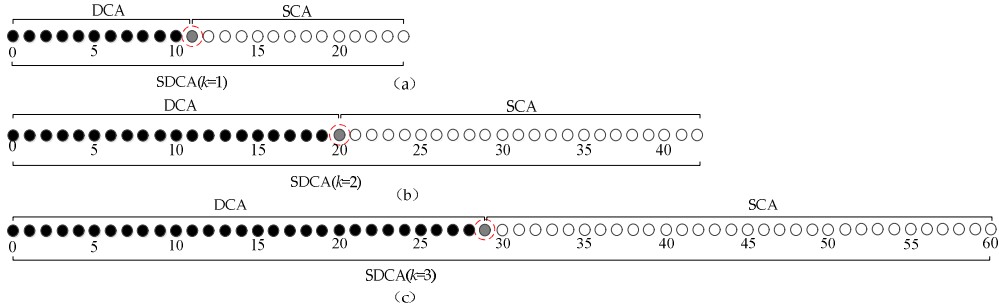

**Figure 3.** The SDCAs of the $k$-level extended array with $N_1 = N_2 = 3$: (**a**) $k = 1$; (**b**) $k = 2$; (**c**) $k = 3$, where black circles, white circles and grey circles, respectively, denote DCA, consecutive parts of SCA, and the overlapping sensors, the overlapping virtual sensors are also marked by red dashed circle boxes.

**Definition 8 (*Coarray Redundancy Ratio, CARR*)**: *For a given sparse array configuration, denote the maximal continuous segments of its DCA and SCA as $(-C_1, C_1)$ and $(-C_3, C_2) \cup (C_2, C_3)$, then CARR is the ratio of the overlapping virtual sensors between the DCA and SCA, which is given by $\varsigma = 2(C_1 - C_2 + 1)/(2C_3 + 1)$.*

**Proposition 2.** *When $N_1 = N_2 = 4$, the CARR of the $k$-level extended sparse array reaches the maximum $\frac{4}{64k+17}$.*

**Proof of Proposition 2.** According to Definition 8, the minimum CARR of the proposed sparse array can be achieved via

$$\begin{aligned} \min \varsigma &= \frac{2N_2 - 4}{4kN_1^2 + 4N_2 + 1} \\ \text{s.t.} N &= kN_1 + N_2 \end{aligned} \tag{13}$$

For the proposed $k$-level extended sparse array, the solution to the above optimal issue can be carried out by using the arithmetic mean-geometric mean (AM-GM) inequalities. Specifically, $\partial \varsigma / \partial N_1 = \frac{-8k^2 N_1^2 - 18k - 16kN_1N_2 + 32kN_1}{\left(4kN_1^2 - 4kN_1 + 4N + 1\right)^2} < 0$ holds due to $N_2 \geq N_1 > 2$; thus, the CARR of the proposed array would decrease monotonically with the increase in $N_1$ and the minimum $\varsigma$ can be obtained when $N_1 = N_2$. Based on the above results, the extremal CARR with respect to $N_1$ and $N_2$ is calculated by

$$\begin{aligned} \max \varsigma &= \frac{2N_1 - 4}{4kN_1^2 + 4N_1 + 1} \\ \text{s.t.} N_1 &= N_2 \end{aligned} \tag{14}$$

By calculating the partial derivative of $\varsigma$ to $N_1$, we have $\partial \varsigma / \partial N_1 |_{N_1=3} = \frac{24k+18}{\left(4kN_1^2 + 4N_1 + 1\right)^2} > 0$, $\partial \varsigma / \partial N_1 |_{N_1=4} = \frac{18}{\left(4kN_1^2 + 4N_1 + 1\right)^2} > 0$ and $\partial \varsigma / \partial N_1 |_{N_1=5} = \frac{-40k+18}{\left(4kN_1^2 + 4N_1 + 1\right)^2} < 0$. This implies that the CARR of the proposed array configuration would increase first (when $3 \leq N_1 = N_2 < 5$) and then decrease with the increase in $N_1$ (when $N_1 = N_2 \geq 5$) and the extremum is $\max\left(\varsigma|_{N_1=4}, \varsigma|_{N_1=5}\right) = \varsigma|_{N_1=4} = \frac{4}{64k+17}$. □

## 4. Simulation Results

In this section, numerical simulations are performed to illustrate the superiority of the proposed *k*-level extended sparse array in terms of array property and DOA estimation performance, where NA, CPA, and DsCAMps are utilized for comparison. It should be mentioned that the array configurations considered here are based on the SDCA to perform a fair comparison. Additionally, all the impinging sources are assumed to have equal power and the corresponding source number is known.

The first simulation compares the CARR of different array configurations and the results are shown in Figure 4, where $k = 2$ is set for the *k*-level extended sparse array. It can be seen that the CARRs of DsCAMps and *k*-level extended sparse array tend to decrease, while that of NA tends to increase with the increase in sensor number. Given the fixed sensor number, the *k*-level extended sparse array exhibits the smallest CARR. Additionally, it should be noted that the maximum consecutive DCA and SCA for the CPA are physically unconnected, which implies that they share no overlapping virtual sensors, and thus the concept of CARR is not applicable to CPA. Furthermore, the influence of *k* on CARR is investigated in Figure 5, and the results show that the three-level extended sparse array has less CARRs than those of two-level extended sparse array and one-level extended sparse array when the sensor number is set to be a certain value.

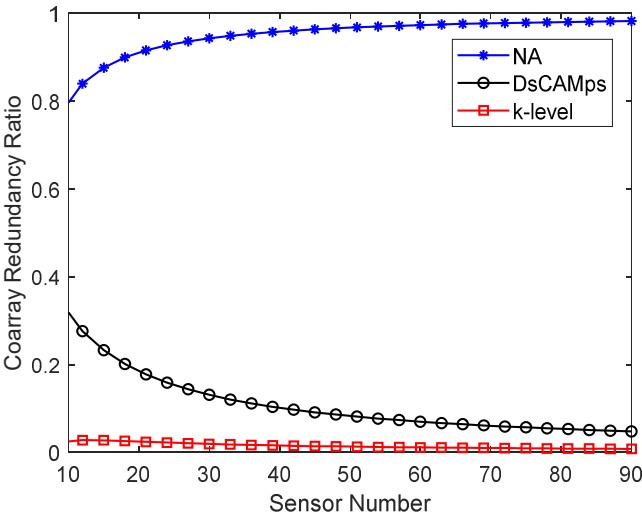

**Figure 4.** The comparisons of CARR.

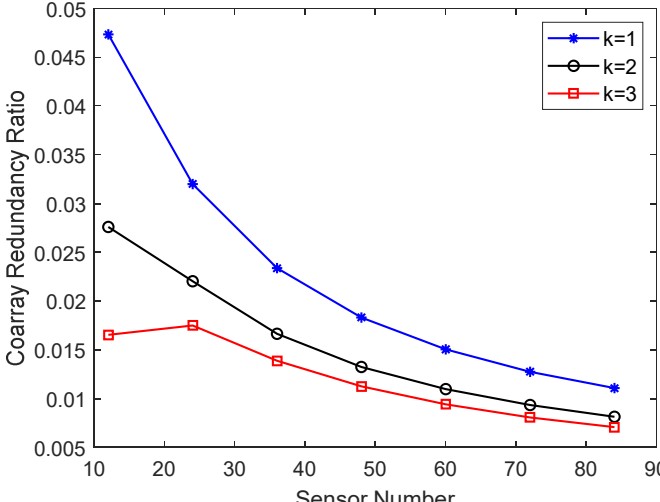

**Figure 5.** The CARR versus *k*.

In the second simulation, the uniform DOF of four array configurations are compared in Figure 6, where $k = 2$ is set for the $k$-level extended sparse array. The results from Figure 6 illustrate that the $k$-level extended sparse array can obtain more uniform DOF than the other array configurations, and the above advantage becomes more evident with the increase in the sensor number. The influence of $k$ on uniform DOF is studied in Figure 7, and the results show that a one-level extended sparse array has more uniform DOF than two-level extended sparse array, and three-level extended sparse array. From the view of array arrangement, three-level extended sparse array has less densely distributed sensors than one-level extended sparse array and two-level extended sparse array, which implies less mutual coupling effects.

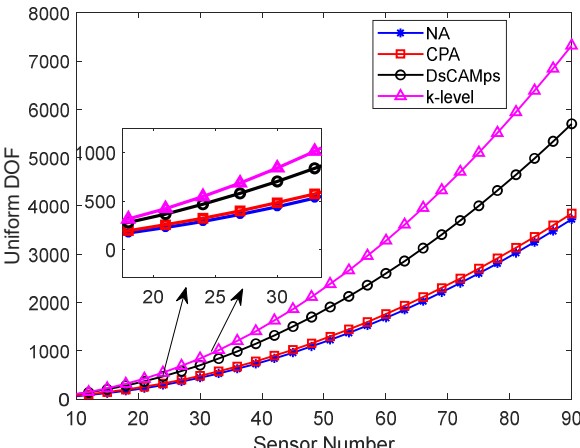

**Figure 6.** The comparisons of uniform DOF.

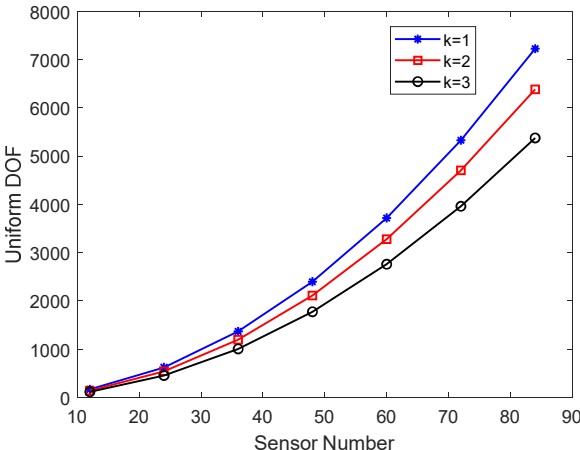

**Figure 7.** The uniform DOF versus $k$.

Then, the DOA estimations of different array configurations are compared by presenting spatial spectra. The number of sensors for all the sparse arrays is uniformly set to 12, and the physical sensor locations are, respectively, given in Table 1. Assume that 16 uncorrelated narrowband sources with uniform distribution between $60°$ and $60°$ impinge on the antenna arrays with $T_p = T_s = 600$ and the SNR being 0 dB. Moreover, the search interval for the spatial spectra is set to be $0.5°$. Figure 8 depicts the MUSIC spectra of four sparse array configurations with $k = 1,2,3$ for the proposed *k-level* extended sparse arrays, where the red-dashed lines and the black-solid lines, respectively, indicate the impinging sources and the estimated spectra. We can observe that although all the sparse arrays can detect 16 sources, the proposed $k$-level extended sparse arrays have more slight bias defined between the peaks and the true DOAs than those of the remaining array configurations, which is mainly attributed to their more uniform DOF and larger effective array aperture.

**Table 1.** Physical sensor locations.

| Array Configuration | Physical Sensor Locations |
|---|---|
| PCA | $\{0, 4, 8, 9, 12, 16, 18, 20, 24, 27, 28, 32\}$ |
| NA | $\{1, 2, 3, 4, 8, 12, 16, 20, 24, 28, 32, 36\}$ |
| DsCAMps | $\{0, 4, 5, 8, 10, 12, 15, 16, 20, 25, 30, 35\}$ |
| 1-level extended sparse array | $\{1, 7, 13, 19, 25, 31, 37, 38, 39, 40, 41, 42\}$ |
| 2-level extended sparse array | $\{1, 5, 9, 13, 17, 21, 25, 29, 33, 34, 35, 36\}$ |
| 3-level extended sparse array | $\{1, 4, 7, 10, 13, 16, 19, 22, 25, 28, 29, 30\}$ |

**Figure 8.** Comparisons of spatial spectra for (**a**) CPA, (**b**) NA, (**c**) DsCAMps, (**d**) 1-level extended sparse array, (**e**) 2-level extended sparse array, and (**f**) 3-level extended sparse array.

The next simulation evaluates DOA estimation performances quantitatively versus SNR and snapshot number by employing the average root mean square error (RMSE), which is defined as

$$RMSE = \sqrt{\frac{1}{QK}\sum_{i=1}^{Q}\sum_{k=1}^{K}\left(\hat{\theta}_{k,i} - \theta_k\right)^2}, \tag{15}$$

where $Q$ denotes the number of Monte Carlo trials, and the $\hat{\theta}_{k,i}$ denotes the estimates of $\theta_k$ for the $i$-th ($1 \leq i \leq Q$) Monte Carlo trial. The array configurations are the same as those in Table 1 and 18 narrow band sources uniformly distributed between $-60°$ and $-60°$ are considered. Figure 9 depicts the RMSE versus SNR with a fixed number of 600 snapshots. It can be observed that, as the number of SNR increases, the RMSEs reduce rapidly for NA, CPA, and DsCAMps until SNR reaches $-2$ dB. In contrast, the RMSEs of $k$-level extended sparse arrays for the cases of $k = 1,2,3$ steadily decreased with the increase in SNR, which are much lower than those of the other three array configurations. Figure 10 plots the RMSE versus the number of snapshots with the fixed SNR 0 dB. The results show that $k$-level extended sparse arrays exhibit a lower RMSE as compared to the NA, CPA, and DsCAMps when the number of snapshots varies from 100 to 1000. The estimation results for $k$-level extended sparse arrays tend to be stable when the number of snapshots reaches about 300, and 600 for the remaining arrays. The above results indicate that the $k$-level extended sparse arrays provide more accurate and stable DOA estimates than the other three-array configurations.

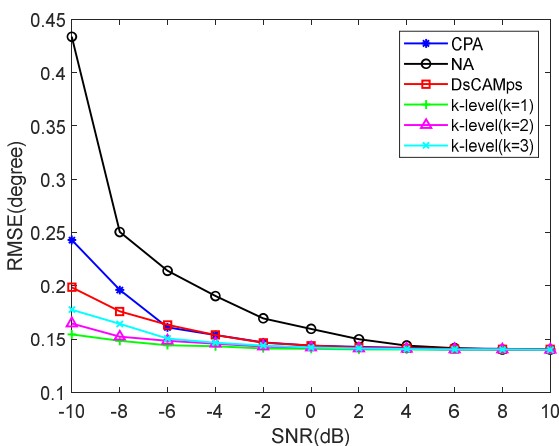

**Figure 9.** RMSE versus SNR with $T_p = T_s = 600$.

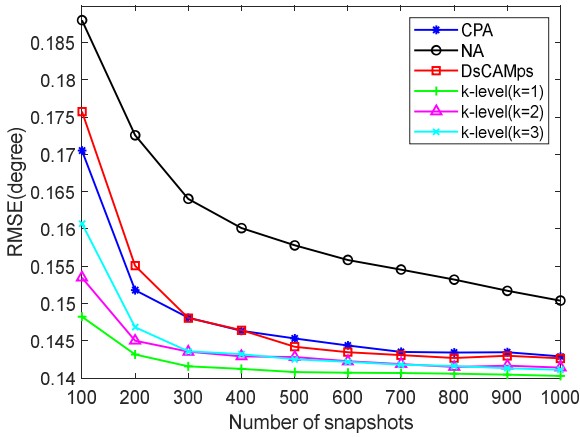

**Figure 10.** RMSE of snapshot number with SNR being 0 dB.

## 5. Conclusions

This paper presents a *k*-level extended sparse configuration from the perspective of sum-difference coarray for DOA estimation, which can provide more uniform DOF and lower coarray redundancy compared to most of the existing sparse array configurations. Then, the closed-form expressions for array geometry and uniform DOF and the corresponding CARR are derived. Based on the *k*-level extended sparse array, the spatial and temporal information of incident sources are jointly exploited for direction-of-arrival estimation. The results of the theoretical analysis and the numerical simulations demonstrate the effectiveness and favorable performance of the proposed ENA in terms of array properties and DOA estimation performance.

**Author Contributions:** Conceptualization, P.Z. and Q.W.; methodology, P.Z. and Q.W.; software, N.W. and G.H.; validation L.W.; writing—original draft preparation, P.Z.; writing—review and editing, P.Z., Q.W., N.W., G.H. and L.W.; visualization, L.W.; project administration, P.Z. and Q.W.; funding acquisition, P.Z., Q.W. and G.H. All authors have read and agreed to the published version of the manuscript.

**Funding:** This research was funded by the National Natural Science Foundation of China under Grant No. 62101223, in part by Key Laboratory of Underwater Acoustic Countermeasure Technology under Grant No. 2022JCJQLB03305, and in part by Natural Science Foundation of the Jiangsu Higher Education Institutions of China under Grant No. 20KJB510027 and No. 20KJA510008.

**Institutional Review Board Statement:** Not applicable.

**Informed Consent Statement:** Not applicable.

**Conflicts of Interest:** The authors declare no conflict of interest.

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
