# Peer review of "k-Level Extended Sparse Array Design for Direction-of-Arrival Estimation"

_electronics, doi:10.3390/electronics11233911_

Round 1

Reviewer 1 Report

- The overall presentation is mathematically sound but generally obscure and the proposed result is difficult to discern among the mathematical derivations. It is strongly suggested that the authors try to clarify the fundamental result and proposed array structure, in order to be useful for the interested reader. 

- In addition, although the whole mathematical construction is clear, a proper theoretical support to provide a better justification of the proposed array remains also unclear. The analysis should be complemented by proper arguments and descriptions that substantiate the proposed array structure.

- The scales in Fig. 8 are not the same across different cases. This fact distorts results and presents a not always fair comparison.

- It is also not clear why higher values of k result in worse error values. A thorough explanation is required in this case.

- There seems to be a lack of comparison in terms of actual run times, although some indicative comparisons are included. A more thorough comparison in terms of complexity is also necessary.  

Author Response

We would like to thank Reviewer 1 for carefully reading our paper and for his/her valuable comments and suggestions for improving the presentation of our paper. We have revised the manuscript carefully and provide the peer-to-peer responses according to the reviewers' comments. Please see included below the list of changes and clarifications that we have made in the revised manuscript. We believe that we have improved the paper accordingly.

Reviewer 2 Report

The paper presents a k-level sparse configuration for DOA estimation. The content covers the expected sections and it is well described the methodology. However, the authors should motivate the paper in the abstract and explain deeply the results, highlighting them wrt the literature

Author Response

We would like to thank Reviewer 2 for carefully reading our paper and for his/her valuable comments and suggestions for improving the presentation of our paper. We have revised the manuscript carefully and provide the peer-to-peer responses according to the reviewers' comments. Please see included below the list of changes and clarifications that we have made in the revised manuscript. We believe that we have improved the paper accordingly.

Reviewer 3 Report

The paper is well written and the proposed material is interesting. I have only the following (very minor) suggestions. The title "Section II" is missing. At rows 121 and 141, remove the word "for" before the symbol "for all". The comma after equation (4) should be replaced by a dot. At row 162, remove "the" immediately after whose.

Author Response

We would like to thank Reviewer 3 for carefully reading our paper and for his/her valuable comments and suggestions for improving the presentation of our paper. We have revised the manuscript carefully and provide the peer-to-peer responses according to the reviewers' comments. Please see included below the list of changes and clarifications that we have made in the revised manuscript. We believe that we have improved the paper accordingly.

Round 2

Reviewer 1 Report

-